# Risk Factors Associated with Children’s Behavior in Dental Clinics: A Cross-Sectional Study

**DOI:** 10.3390/children11060677

**Published:** 2024-06-03

**Authors:** Rana Abdullah Alamoudi, Nada Bamashmous, Nuha Hamdi Albeladi, Heba Jafar Sabbagh

**Affiliations:** 1Pediatric Dentistry Department, Faculty of Dentistry, King Abdulaziz University, Jeddah 21589, Saudi Arabia; rasalamoudi@kau.edu.sa (R.A.A.); nobamashmous@kau.edu.sa (N.B.); 2General Dentistry, King Abdulaziz University Dental Hospital, Jeddah 22252, Saudi Arabia; nhhalbeladi@kau.edu.sa

**Keywords:** behavior, children, dental, risk factors

## Abstract

One of the most crucial tasks of pediatric dentists is to control children’s negative behaviors. This study aimed to assess dental behavior and the associated risk factors among children aged 4–12. This cross-sectional study recruited healthy, unaffected children aged 4 to 12 years. Parents were interviewed regarding the sociodemographic details and characteristics of their children’s dental visits. Two collaborative dentists examined the children for dental caries (DMFT/DMFT) and behavioral status (Frankl’s behavior rating scale). This study included 439 children: 27.3% exhibited uncooperative behavior, and the mean DMFT/dmft was 8.46 ± 3.530. Uncooperative behavior significantly increased when the dental visit was scheduled as an emergency treatment (*p* = 0.134; Adjusted Odds Ratio (AOR): 1.530) and when there was an elevated DMFT/DMFT ratio (*p* < 0.001; AOR: 1.308). This study revealed a significant association between children’s uncooperative behavior and their first dental visit, emphasizing the need for tailored strategies to address behavioral challenges when scheduling pediatric dental care. The proactive measures included controlling caries and avoiding emergencies.

## 1. Introduction

One of the most crucial tasks for pediatric dentists is to evaluate children based on their dental behavior because of behavior management problems and associated outcomes. Thus, studies have been conducted to assess children’s behavior in different dental settings [1]. Some children handle their visits successfully, whereas others display negative or uncooperative behaviors, referred to as dental behavior management problems, in which case therapy may be terminated or interrupted, leading to poor oral health [2].

Predicting a child’s behavior before their first dental appointment helps dentists direct the child’s behavior while receiving dental care [3]. Studying and analyzing individual factors is challenging because of the interplay of several variables frequently proposed as components of a child’s behavioral patterns. Researchers are still attempting to determine the factors that should be focused on to improve children’s behavior in the dental environment. While no assessment technique or approach can accurately predict a child’s behavior during dental treatment, understanding the various background elements that influence a child’s behavior can help predict pediatric patient behavior [4].

Individual, parental, and environmental factors can be used to categorize determinant variables and concomitant risk factors for dental caries and associated behaviors among children [5,6]. Children’s dental visit behaviors are also influenced by their characteristics, such as dental experiences [7] and developmental and cognitive traits [8,9]. Parental variables directly linked to children’s oral health include sociodemographic variables, behaviors related to oral health, anxiety, cognition, and other attributes. The family environment also shapes children’s oral health behavior; thus, it is necessary to thoroughly assess dental caries and the concomitant clinical circumstances in children, as well as the related familial factors [8,9]. There is a correlation between parental education level and income and children’s dental behavior and anxiety. According to several studies, children from high-income families, who also have lower rates of dental caries, exhibit more cooperative behaviors [10,11]. Furthermore, a 2011 cohort study suggested that previous experience significantly affected children’s dental behavior [12]. Nevertheless, a few recent studies have examined children’s behavior in dental clinics and the related risk factors in China, as Geo et al. (2021) reported, highlighting the need for further research worldwide [13].

Although most Saudi Arabian children suffer from oral diseases [7,14,15], little is known about the risk factors responsible for their behavior in dental clinics. As a result, current epidemiological information is required regarding the risk factors associated with a child’s dental behavior. Therefore, this study aimed to evaluate the dental behavioral status of children aged 4–12 years visiting pediatric dental clinics in a university hospital and the related factors in Jeddah, Saudi Arabia.

## 2. Materials and Methods

This cross-sectional study recruited healthy children aged 4 to 12 years from the University Dental Hospital (UDH) in Jeddah, Saudi Arabia, in 2022. This research (No. 336-11-21) was approved by the Research Ethics Committee of King Abdulaziz University (KAU). The inclusion criterion was healthy children (ASA1) [15] treated at pediatric dental clinics. The participants were enrolled from regular and emergency pediatric dental clinics to ensure the inclusion of children exposed to various dental environments. The sample size was calculated using the Collaborative Open-Source Project in Epidemiologic Computing (OpedEpi) version 3.01, following the methodology outlined by Alabdullatif et al. (2023) [5]. A total of 292 children were needed for a sample power of 80%, an odds ratio of 1.99, a difference of 17% between children with uncooperative and cooperative behaviors, exposed or not exposed to a previous dental treatment, and a 95% confidence interval.

The parents were provided with a consent form containing the research details, a confidentiality statement, and a questionnaire. Subsequently, two collaborative dentists interviewed the parents. After the interview, the children were examined by dentists, and their teeth were polished using a prophylaxis paste.

The data collection form included the following:

General information, including sociodemographic details such as the child’s age, parental education, and parental status, and family income was categorized as low (less than 7000 SAR), moderate (7000–12,000 SAR), or high (more than 12,000 SAR).Dental visit characteristics, comprising information on dental visit characteristics, were enquired upon, including the type of pediatric dental clinic (emergency vs. regular pediatric dental clinic), whether this had been the first dental visit (yes/no), and whether the child had been treated in a regular or emergency pediatric dental clinic in the previous two years (yes/no).The WHO dental caries status (DMFT/dmft) was calculated by summing the scores for decayed (D), missing (M) teeth due to caries, and filled (F) teeth for each participant’s primary and permanent teeth [16].Frankl’s behavior-rating scale categorizes children into four groups based on their attitudes toward dental treatment: Rating 1 (definitely negative), Rating 2 (negative), Rating 3 (positive), and Rating 4 (definitely positive). For this analysis, the participants’ behaviors were dichotomized as uncooperative (ratings 1 and 2) or cooperative (ratings 3 and 4) [4].

Two collaborative dentists were responsible for data collection. An inter-reliability test was conducted on ten children, resulting in a Kappa score of 0.92 (near-perfect agreement). Additionally, ascertainment was confirmed through several meetings and discussions of questionnaire answers and behavior-rating criteria.

### Statistical Analysis

Descriptive statistics are presented as frequencies and percentages for categorical variables and means and standard deviations (SD) for continuous variables. Comparisons were conducted using chi-square tests for the nominal variables, t-tests for the continuous variables, and independent sample Mann–Whitney U tests for the nonparametric data. Binary regression analysis tested the associations between ER-C treatment (dependent variable) and sociodemographic factors, DMFT/DMFT, behavior at dental clinics, dental neglect, and dental barriers (independent factors). The significance level was set to *p* < 0.05.

## 3. Results

In this study, 439 children were included, 45.8% of whom were male and 54.2% of whom were female. The sample comprised 207 (47.2) children from emergency clinics and 232 (52.8) from regular pediatric dental clinics. In total, 27.3% of the participants exhibited uncooperative behaviors during their dental visits, while 72.7% were cooperative. The mean age of the children was 7.82 ± 2.081, and the mean DMFT/dmft was 8.46 ± 3.530 (Table 1).

Table 1 illustrates the sample’s distribution based on the sociodemographic variables, the details of prior dental visits, the DMFT/DMFT, and the cooperation levels during dental treatment. Notably, a significantly higher proportion of uncooperative children came from emergency dental clinics (60.8%) than from regular pediatric dental clinics (39.2%) (*p* < 0.001). Furthermore, children with a lower socioeconomic status demonstrated a higher tendency for uncooperative behavior, as evidenced by significant associations with fathers’ education (*p* < 0.001), mothers’ education (*p* = 0.015), and family income (*p* = 0.012).

Table 2 presents the regression analysis’ results. This study found that a child’s uncooperative behavior significantly increased during the first dental visit (AOR: 2.852; *p* = 0.004) and when there was an elevated DMFT/dmft (*p* < 0.001; AOR: 1.308). Conversely, an older age was associated with a significantly less uncooperative behavior (AOR: 1.308; *p* < 0.001). Although not statistically significant, emergency treatment increased the AOR for uncooperative behavior (*p* = 0.134; AOR: 1.530).

## 4. Discussion

This study examined the risk factors related to children’s behavior during emergency visits and regular dental clinic visits at UDH in Jeddah. In Saudi Arabia, very few studies have assessed children’s dental behavior risk factors, stressing the type of pediatric dental clinic [5,8,17,18]. Our study’s mean dmft score (8.46 ± 3.530) aligned with prior studies conducted on Southeast Asian children [19]. However, this was greater than what had been previously reported in Saudi children, ranging from 5 to 7 [16,20,21]. This could be due to the different targeted groups, which did not include emergency clinics in previous studies.

According to this study, children who attended emergency dental clinics rather than regular dental clinics were more likely to display uncooperative dental behaviors. This finding guides dentists in the decision to treat children in emergency clinics. Additionally, it highlights the importance of behavioral guidance and the application of various behavioral management techniques when treating emergency cases [22].

Our investigation was conducted on a population aged 4–12 years to determine whether there was a relationship between sociodemographic variables and the behavior of children at dental clinics and prior dental visits. A significant amount of research has shown that the prevalence of dental caries is inversely correlated with one’s socioeconomic level [23,24,25]. According to this study’s analysis of the socioeconomic factors, the degree of parents’ education was substantially associated with children’s uncooperative behaviors toward dental caries. These findings concur with research showing a relationship between parents’ socioeconomic status and their children’s dental behavior [26].

This study’s collected data indicated a correlation between children’s uncooperative behavior, dental caries, and low family income. Children from low-income families frequently consume diets high in sugar and fat and lack nutrients, which predispose them to the risk of developing obesity and dental caries [27]. A diet high in sugar-filled foods and beverages, accounting for more than 10% of the total daily dietary energy intake, i.e., 18.25 kg/person/year, is closely linked to a general rise in caries’ prevalence and indices in communities. Because of this, the World Health Organization (WHO) guidelines advise consuming no more sugar than the abovementioned level or, preferably, no more than 5%, i.e., 9.08 kg/person/year of cumulative dietary energy [28]. This increasing use of sugar, which is harmful to dental health, is especially prevalent among children of immigrant communities (from industrialized and developing nations), whose customs and beliefs are altered via an inevitable process of acculturation [29].

A low socioeconomic status is associated with fewer dental visits and reduced caries’ prevention and treatment [30]. Furthermore, when sociodemographic factors were considered in our analysis, children’s dental treatment at an emergency clinic was not linked to uncooperative behaviors. This suggests that children’s caries experience was unaffected by the dental attendance within the low sociodemographic group. This could be explained by the widespread practice among parents of only taking their children to the dentist when they have a toothache or at least one cavity, thus skipping routine oral examinations and preventative measures [31]. Although prevention is successful in dentistry, there is little evidence that dentists understand or use it in their daily clinical work [32]. The literature and our study’s findings on the relationship between a child’s behavior with caries and their family’s socioeconomic level should motivate the development of more potent preventative methods targeted explicitly at lower socioeconomic classes.

In our investigation, 20.5% of the patients visited an emergency dental clinic first. Among these first visitors, the percentage of cooperative children was lower (10.7%) than that of uncooperative children (46.7%), showing a statistically significant difference (*p* < 0.001). This percentage was lower than an identical profile analysis of children who presented to a university pediatric dental clinic for dental emergencies in Belgium [33]. Approximately 47% of the patients with UDH had experienced emergency dental appointments. This is noteworthy, as the cohort only presented themselves to a clinic when there was a problem, even though they were aware of and had access to normal oral health treatments. Children’s oral and general health suffers as a result of this problem-focused and delayed care-seeking behavior, because they experience persistent discomfort and ultimately need more intrusive procedures which could have been prevented [34].

Furthermore, a regression analysis was performed, which modified the impact of extraneous variables to ascertain the objective link between them. When the variables were matched, older children exhibited a lower propensity to receive therapy at emergency dental clinics. This study found a significant correlation between age and dental behavior. This indicates that a child’s ability to follow dental care plans improves with age in the 4–12 age range. This finding aligns with previous research conducted in Edirne, Turkey, where cooperative behavior in children increased with age [35]. Over time, treatment factors and subjective experiences appear to have less impact on dental fear in children. Dentists and patients face challenging circumstances when a child requires emergency care [36]. However, our findings contradict those of another survey conducted in Saudi Arabia, where older children displayed a greater tendency for uncooperative behavior at an emergency dental clinic [15]. Current research suggests that dentists should consider the importance of conditioning and progressive exposure to obtain a child’s cooperation during dental treatment [37]. However, sex was not significantly associated with dental behavior.

This study focused on the importance of planning a child’s first visit to the dentist, which is a critical step for their mental health and behavior in the dental clinic. Moreover, it is crucial to prepare parents beforehand, as parental style and attitude affect children’s behavior [38]. Avoiding invasive or emergency treatments during the first visit is also essential. Emergency treatment has been found to increase the likelihood of uncooperative behavior in children. Although the relationship was not significant, a previous study reported an increase in negative behavior among pediatric patients exposed to invasive procedures such as extractions and pulpectomies [39]. Additionally, patients needing emergency treatment often experience pain, which tends to increase uncooperative behavior [40].

This study has several strengths. No other research has evaluated children’s behavior and assisted them in dental clinic appointments. This study offers an update on these parameters. Biases related to recall and desirability were observed. Because non-probability sampling was used, it was impossible to extrapolate the results to the entire community. However, the research sample was drawn from a university hospital that treats a diverse range of patients, including those from various nations.

Furthermore, the gender and socioeconomic status distributions of the controls matched those of the general population, resolving generalization-related concerns. The extensive age range of the participants in this study is another drawback, because it may have acted as a confounding factor and altered the findings. Nevertheless, by recognizing and addressing children’s dental anxiety and behavioral risk factors, dental professionals can enhance children’s overall experience and promote better oral health outcomes. Further research and the development of targeted interventions are warranted to effectively manage pediatric dental behavior and improve the quality of care provided to young patients.

## 5. Conclusions

This study revealed a significant association between children’s uncooperative behavior and their first dental visit, emphasizing the need for tailored strategies to address behavioral challenges when scheduling pediatric dental care. The proactive measures included controlling caries and avoiding emergencies.

## Figures and Tables

**Table 1 children-11-00677-t001:** Distribution of subjects according to their sociodemographic variables and clinical dental visit characteristics (N = 439).

Variable		Child Behavior		Total	*p* Value
		Uncooperative (%)	Cooperative (%)		
**Sociodemographic variables**					
**Child age**	**Mean ± SD**	6.01 ± 2.410	8.66 ± 2.410	7.82 ± 2.081	<0.001 *^t^
**Child gender**	Male	52 (43.3)	149 (46.7)	201 (45.8)	0.527 ^c^
	Female	68 (56.7)	170 (53.3)	238 (54.2)	
**The child lives with…**	Single parents	15 (12.5)	33 (10.3)	48 (10.9)	0.519 ^c^
	Both parents	105 (87.5)	286 (89.7)	391 (89.1)	
**Family income**	Less than 7000 SAR	42 (35.0)	68 (21.3)	110 (25.1)	0.012 *^c^
	SAR 7000–12,000	53 (44.2)	166 (52.0)	219 (49.9)	
	More than 12,000	25 (20.8)	85 (26.6)	110 (25.1)	
**Father education**	High school or less	40 (33.3)	60 (18.8)	100 (22.8)	0.001 *^c^
	More than high school	80 (66.7)	259 (81.2)	339 (77.2)	
**Mother education**	High school or less	79 (65.8)	169 (53.0)	248 (56.5)	0.015 *^c^
	More than high school	41 (34.2)	150 (47.0)	191 (43.5)	
**Dental visit characteristics**					
**Is this your child’s first visit?**	Yes	56 (46.7)	34 (10.7)	90 (20.5)	<0.001 *^c^
	No	64 (53.3)	285 (89.3)	349 (79.5)	
**Type of dental clinic**	Emergency	73 (60.8)	134 (42.0)	207 (47.2)	<0.001 *^c^
	Regular	47 (39.2)	185 (58.0)	232 (52.8)	
**Child visited dental emergency clinic in the last 2 years**	Yes	43 (35.8)	133 (41.7)	176 (40.1)	0.264 ^c^
	No	77 (64.2)	186 (58.3)	263 (59.9)	
**Child visited regular dental clinic in the last 2 years**	Yes	49 (40.8)	236 (74.0)	285 (64.9)	<0.001 *
	No	71 (59.2)	83 (26.0)	154 (35.1)	
**Clinical examination**					
**DMFT + dmft**	**Mean ± SD**	9.43 ± 3.483	8.09 ± 3.483	8.46 ± 3.530	<0.001 *^m^

* Significant level at *p* = 0.05, ^m^ Mann–Whitney U test; ^t^ *t*-test; and ^c^ chi-square for categorical variables.

**Table 2 children-11-00677-t002:** Regression analysis for the relationship between child behavior (dependent factor) and model 1 (sociodemographic data and dental visit type (independent factor)).

Variable		AOR (95% CI) *p* Value
**Child gender**	Male	0.589 (0.337–1.029) 0.063
	Female	1.000
**Father education**	High school or less	1.423 (0.678–2.987) 0.351
	More than high school	1.000
**Family income**	Less than 7000 SAR	1.132 (0.468–2.740) 0.784
	SAR 7000–12,000	0.823 (0.414–1.638) 0.580
	More than 12,000 SAR	1.000
**Emergency**	Yes	1.530 (0.878–2.669) 0.134
	No	1.000
**Is this your child’s first visit?**	Yes	2.852 (1.400–5.810) 0.004
	No	1.000
**The child lives with…**	Single parents	0.624 (0.264–1.478) 0.284
	Both parents	1.000
**Child age**		1.308 (1.190–1.439) <0.001
**DMFT + dmft**		1.308 (1.190–1.439) <0.001

## Data Availability

The data used in this study are available upon reasonable request from the corresponding author. The data are not publicly available due to ethical consideration.

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
