# Peer review of "Risk Factors Associated with Children’s Behavior in Dental Clinics: A Cross-Sectional Study"

_children, 2024, doi:10.3390/children11060677_

Round 1
Reviewer 1 Report
Comments and Suggestions for Authors
With interest I’ve read the paper “Risk factors associated with child’s behavior in the dental clinics. A cross sectional study”. The authors have chosen an important topic and assessed dental behavior using Frankl’s behavior rating scale
and caries status among children aged 4 to 12 years.
The article is interesting, however, some important comments should be addressed.
General note: the text should be proofread by a native speaker.
Abstract
There is no need to write (1) Background:, (2) Methods:, and etc.
From the abstract it seems that DMFT and behaviour were the only things assessed. The questionnaire is not reflected at all.
Line 21 “Uncooperative behavior significantly increased when the dental visit 21 was scheduled as an emergency treatment (P=0.134…” This p shown insignificance of differences!
Introduction
Line 41 “Researchers…” this sentence needs referencing.
Lines 46-50 Please, provide references
The information is lacking regarding some previous studies, assessing children’s behavior during dental visits.
Materials and methods
Line 65. What is meant by healthy unaffected children?
Line 75. What is meant by cases and what is meant by controls?
Results
The tables are not formatted according to the recommendations in authors guidelines.
Discussion, conclusion - no comments
Comments on the Quality of English Language
The text should be proofread by a native speaker for style and grammar correction.
Reviewer 2 Report
Comments and Suggestions for Authors
Results from regression analysis were interesting. Why only father's education mattered but not mother's education?
The odds ratios were adjusted odds ratios. Please explain in the Methods what is meant by adjusted.
"The study found that a child's uncooperative behavior significantly increased when the dental visit was scheduled as an emergency treatment (P=0.134; Adjusted Odds Ratio (AOR): 1.530)". This point is intuitive, but not justified in this study as the P value is very large (0.134). The regression model in Table 2 showed that 1st visit, child age, and DMFT+dmft were the only 3 significant factors in influencing child uncooperative behavior. That means emergency visit vs regular visit was not an important factor. The Conclusion and Abstract conclusion are, therefore, misleading and should be revised to reflect what the data said.
Statistical analysis section mentioned that the authors used Independent-Samples Mann-Whitney U Test for nonparametric data. Please indicate in the Results section where this Mann-Whitney U test was applied.
Round 2
Reviewer 2 Report
Comments and Suggestions for Authors
The authors have satisfactorily addressed my concerns.